# Comprehensive Characterization and In Vitro Functionality Study of Small Extracellular Vesicles Isolated by Different Purification Methods from Mesenchymal Stem Cell Cultures

**DOI:** 10.3390/ijms262110602

**Published:** 2025-10-30

**Authors:** Marta Venturella, Ali Navaei, Davide Zocco

**Affiliations:** 1Lonza, 53100 Siena, Italy; marta.venturella@lonza.com; 2Lonza, Houston, TX 77047, USA; ali.navaei@lonza.com

**Keywords:** mesenchymal stem cells (MSCs), small extracellular vesicles (sEVs), cell-free therapy, 2D cell culture, 3D cell culture, bioreactor, in vitro cell-based assays

## Abstract

Mesenchymal stem cells (MSCs) exhibit therapeutic properties, which have been attributed to their secretome, the set of secreted factors comprising cytokines, growth factors, and extracellular vesicles (EVs). In particular, small extracellular vesicles (sEVs) or exosomes, ranging between 30 nm and 120 nm in diameter, can target specific tissues to deliver molecular payloads, thus lending themselves as promising platform for cell-free therapies. In this study, sEVs were purified from the conditioned medium (CM) harvested from human bone marrow-derived MSC culture and purified using size-exclusion chromatography (SEC) or density gradient ultracentrifugation (DG-UC). Then sEVs were fully characterized for identity and integrity using multiple analytical methods, including single-particle, transcriptomic and proteomic analyses. Different in vitro cell-based assays were established to evaluate the biological effects of the purified sEVs. Specifically, scratch wound healing and tube formation assays using human umbilical vein endothelial cells (HUVECs) were used to evaluate the regenerative properties of MSC-sEVs. Our findings demonstrated that the in vitro functional properties of MSC-sEVs are correlated with sEVs’ purity levels obtained by different purification methods.

## 1. Introduction

Among stem cells, mesenchymal stem/stromal cells (MSCs) have gained significant attention in the field of regenerative medicine, as they appear to be safe and effective according to clinical data collected over the last decades [1,2,3,4,5]. MSCs are multipotent, non-hematopoietic adult stem cells, typically found in a variety of human tissues, including bone marrow, adipose tissue, liver, intestine, lung, connective muscle tissue, spleen, skin, placenta, umbilical cords, and other tissues [6]. MSCs are characterized by abilities of self-renewal, differentiation, immunomodulatory, and trophic support [7]. MSCs exhibit the “trilineage potential” of osteogenic, chondrogenic, and adipogenic differentiation capabilities in vitro [8]. The effects of MSCs are mediated by paracrine signaling, including their secreted soluble factors (e.g., growth factors, cytokines, and chemokines) and extracellular vesicles (EVs) [9,10]. In particular, small EVs (sEVs) or exosomes have been reported to be therapeutically efficacious in various preclinical models by virtue of their intrinsic cargo [11,12,13].

Development of cell-free therapies is important because only less than 1% of transplanted MSCs reach the injured sites and the rest are trapped in the liver, spleen, and lung after administration. Moreover, administration of cell therapy is not free from safety risks, including cellular rejection, toxicity, and tumorigenic potential [14,15]. In this regard, MSC-EVs represent a legitimate alternative to stem cells, as they hold the therapeutic potential of the cell source while lacking self-propagating properties, and they can be easily sterilized by filtration. MSC-EVs have been applied as new cell-free therapies in a variety of disease models including neurological, cardiovascular, immune, renal, musculoskeletal, liver, respiratory, eye, and skin diseases, as well as cancers [1,9,16,17]. One of the most important effects of MSC-EVs is their pro-angiogenic capacity [18,19]. In fact, vascularization is required for tissue repair, since it provides a sufficient supply of oxygen, nutrients and growth factors. Moreover, MSC-EVs can be engineered to transport and deliver therapeutic cargo to target diseased cells and tissues [8,15].

In this study, we assessed the impact of different purification methods on sEVs’ purity and their biological effects on in vitro models. Specifically, sEVs were purified from human bone marrow MSC-conditioned medium (CM; from both 2D and 3D cultures) with two EV purification methods: density gradient ultracentrifugation (DG-UC), considered the gold standard method for isolating pure EVs, and size-exclusion liquid chromatography (SEC), a convenient and fast technique commonly used to purify intact EVs. It is known that the choice of EV isolation technique has an impact on the grade of purity of EVs and, consequently, on their functional effects [11,20]. DG-UC should result in a final EV sample with high purity, but the workflow is laborious and time-consuming (i.e., not manufacturing-friendly). On the contrary, SEC is a relatively faster and easier method, which allows the users to reach a medium to high EV purity, but the final sample is diluted, and it requires further processing.

The two sEV sample preparation methods were fully characterized with various analytical methods, such as single-particle analysis, Transmission Electron Microscopy (TEM), and proteomic and transcriptomic platforms. Then, in vitro cell-based assays were established using human umbilical vein endothelial cells (HUVECs) to evaluate the biological effects of the two sEV samples. HUVECs are a cell type widely used for cell proliferation, wound healing, and tube formation assays and MSC-EVs can be taken up by endothelial cells and stimulate their proliferation, differentiation, and migration [21,22].

As expected, sEVs purified by DG-UC had the highest purity index, but both sEV samples were comparable in terms of particle integrity and expression of canonical exosome markers. Moreover, both sEV samples exerted functional effects on endothelial cells compared with the untreated control, but, as expected, the sample with highest purity, sEVs purified by DG-UC, exerted biological effects at a lower dose.

## 2. Results

### 2.1. Purification of MSC-EVs

Concentrated CM was purified by two methods: DG-UC and SEC. Purification runs were carried out in triplicate. Ten fractions of 1 mL were recovered by DG-UC. Particle concentration (Particles/mL) was measured in each fraction using a Flow NanoAnalyzer, while EV identity was assessed by the presence of the three tetraspanins CD9, CD63, and CD81 with an ELISA assay. Figure 1 shows an example of the overlay of results of the two assays for one purification run.

Fractions 7 and 8 were selected, as they had the highest particle concentration and tetraspanin signal determined by ELISA. Then, the two fractions were stained with CFSE to evaluate their EV integrity, and with a mix of three fluorescent antibodies against CD9, CD63, and CD81 to quantitatively assess the expression of these EV biomarkers. 5(6)-CFDA-SE (CFSE) is a non-fluorescent molecule carrying two acetate groups that facilitate transit through the EV membrane. Once inside the EVs, it is converted into fluorescent CFSE by intracellular esterases removing the acetate groups. CFSE-positive particles are intact vesicles that contain cellular esterases.

A higher percentage of staining with CFSE was obtained for Fraction 7 compared with Fraction 8 (64.7% for Fraction 7 and 32.2% for Fraction 8). Moreover, Fraction 8 was characterized by a higher noise threshold value of the FITC channel (206 compared with 21 for Fraction 7; 15–17 for unstained samples). The higher threshold may be due to excess of unpurified dye and related to the presence of protein contaminants. In addition, a higher percentage of staining with the mix of tetraspanin antibodies was obtained for Fraction 7 compared with Fraction 8 (68.7% for Fraction 7 compared with 51.7% for Fraction 8). On the basis of the abovementioned results, Fraction 7 was identified as the fraction with the highest percentage of intact sEVs and with the lowest amount of impurities. Hence, Fraction 7 samples from two independent DG-UC purifications were pooled, washed with PBS, and further purified by ultracentrifugation to eliminate iodixanol from the final sEV sample.

Figure 2 shows a representation of one SEC elution profile, obtained following the procedure described in the Materials and Methods (Section 4.3). To identify the right fractions, we plotted particle concentration, EV identity assessed by the CD9, CD63, and CD81 ELISA assay, and protein concentration measured by the microBCA assay. Fractions with a high particle concentration (orange line) and that were positive for the three tetraspanins CD9, CD63, and CD81 (blue line), but with a low protein concentration (green line), were pooled and concentrated to a final volume of 0.5 mL using a 100 kDa ultrafiltration device. Small protein aggregates have sizes comparable with small EVs and could not be completely removed from the purified sample.

The described workflow was followed for both the two sources of CM (2D and 3D), and the purification runs were carried out in triplicate.

### 2.2. Assessment of MSC-EVs’ Purity, Integrity, and Identity

First, MSC-EVs were characterized with a Flow NanoAnalyzer to determine the particle concentration and size distribution. The instrument is calibrated with standard size beads, which allows the characterization of a population of sEVs (40–200 nm). Consistent with the existing literature [23,24,25], the average particle concentrations from 3D cultures were higher than those from 2D cultures, although the differences were not statistically significant. SEC-purified samples from 2D cultures had higher particle concentrations than DG-UC-purified ones, though the difference was not statistically significant. Mean sEV sizes were similar across 2D and 3D cultures and purification methods (Figure 3).

The purity index was calculated via dividing the mean particle concentration by the mean total protein concentration (microBCA assay) [20]. As expected, the DG-UC method yielded the highest purity index (Figure 4).

To quantify the percentage of intact EVs, purified sEVs were then stained with CFSE. To reduce background noise, the excess of dye was removed with SEC prior to analysis with Flow NanoAnalyzer. All samples showed a percentage of CFSE staining above 60%: 2D-sEVs purified by SEC, 74.3%; purified by DG-UC, 64.7%; 3D-sEVs purified by SEC, 70.2%; purified by DG-UC, 70.5%. EV integrity was also qualitatively investigated by Transmission Electron Microscopy (TEM). TEM confirmed the presence of lipid bilayer-surrounded vesicles in all sEV samples. Figure 5 shows examples of TEM images of MSC-sEVs derived from 2D cultures and 3D cultures, purified by SEC chromatography (A,C) and DG-UC (B,D). For both sources of CM, the pictures of samples purified by the two methods show a similar size and integrity of sEVs. Compared with the 3D samples, 2D samples show a lower particle concentration, as expected from the data obtained by the Flow NanoAnalyzer, and different sizes of EVs (a population of MVs (microvesicles) of 200 nm in size and a population of sEVs).

Samples were characterized also in terms of EV identity, using two orthogonal methods (Flow NanoAnalyzer and Western blotting). Samples were stained with a mix of three fluorescent antibodies targeting tetraspanins (CD9, CD63, and CD81) and, after the removal of unbound antibodies, analyzed with a Flow NanoAnalyzer. Tetraspanin-positive percentages were higher in samples derived by DG-UC, compared with those from SEC: 2D sample purified by SEC, 47.4%; purified by DG-UC, 68.7%; 3D sample purified by SEC, 44.1%; purified by DG-UC, 67.5%. DG-UC samples showed higher percentages compared with SEC samples, and this can be explained by the higher purity index of these sEVs (Figure 4).

Single staining of sEV samples with fluorescent antibodies against each tetraspanin were also performed, but only for 3D culture. Figure 6 shows the percentage of staining for each tetraspanin in SEC- and DG-UC-derived sEVs. Similar expression patterns were found in the two sets of samples, with CD63 and CD9 being the most and the least expressed tetraspanins, respectively.

SEC- and DG-UC-derived sEVs were analyzed by Western blotting, loading the same number of particles, to confirm the presence of canonical small EV markers (CD81 and Alix) and the absence of the non-EV marker (calnexin). The MSC cell lysate was loaded as positive control for calnexin blotting. The obtained results confirmed the expression of sEV markers in both sEV preparations, and the absence of calnexin expression (Figure 7).

### 2.3. Characterization of MSC-sEV Cargo

We performed an extensive characterization of the total protein and miRNA cargo present in the two sEV preparations, limited to 2D samples. This analysis provides complementary information about the EVs’ identity and payload. EV-related markers are assessed among the total items detected, based on public databases, e.g., ExoCarta. Moreover, a comparative analysis between the two EV purification methods provides a measure of similarity between the two methods, providing a complementary parameter in addition to the purity index. Finally, we analyzed the over-expression of markers which are associated with the biological functions of EVs, such as cell proliferation, cell differentiation, and angiogenesis.

#### 2.3.1. Proteomic Analysis

MSC-derived sEVs purified by SEC and DG-UC from 2D CM were further characterized for total protein cargo. Starting from the same amount of total proteins, mass spectrometry was performed to determine the proteomic signature of the two sEV purification samples in duplicate.

Protein hits were identified using both one-peptide and two-peptide thresholds. Using the two-peptide threshold, which is more stringent, 150 proteins were identified in SEC-sEVs and 135 proteins in DG-UC-sEVs. The Jaccard coefficient was 55.37% (Figure 8A). If we consider only EV-related proteins, 95 proteins were found in SEC-sEVs and 93 proteins in DG-UC-sEVs. The Jaccard coefficient was 56.67% (Figure 8B).

Due to the intrinsic limitations of the SEC purification method, which cannot eliminate the presence of co-purified proteins, EV-associated proteins, or protein aggregates present in CM within the same size range of sEVs, it is expected to find more co-isolated proteins in SEC-derived sEVs, compared with DG-UC-sEVs. However, if we consider only EV-related proteins, the two samples are closer to each other. The similarity index values are considered medium to high, due to the intrinsic variability of the analytical technique, so running replicates of the same sample should result in a Jaccard coefficient with a maximum of 70–80% [26]. In fact, less abundant proteins are close to the detection limit and their identification could be challenging, depending on the run. The results obtained using the one-peptide threshold are very similar to those reported above (Appendix A).

Protein similarity between the two samples (purified using either SEC or DG-UC) was also calculated using Spearman’s correlation coefficient which was 0.76, indicating medium to high correlation (Figure 9).

Considering also proteins identified by the one-peptide method, Cellular Component (CC) Gene Ontology (GO) analysis was carried out for the proteins commonly detected in SEC-sEVs and DG-UC-sEVs (113 proteins). Proteins were significantly enriched in the “Extracellular” and “Exosomes” GO annotations (Figure 10), confirming the EV identity of the two samples.

Biological Process and Biological Pathways GO analyses are reported in Appendix A. Among the common proteins identified in SEC- and DG-UC-sEVs, only a few proteins are up-regulated in SEC-sEVs and DG-UC-sEVs (Figure 11), which suggests the similarity of the two sEV samples to each other. A detailed list of proteins which are up-regulated in SEC-sEVs compared with DG-UC-sEVs (blue dots in the volcano plot) and up-regulated in DG-UC-sEVs compared with SEC-sEVs (orange dots in the volcano plot) is reported in Appendix A.

#### 2.3.2. Transcriptomic Analysis

MSC-derived sEVs purified from 2D CM by SEC and DG-UC were characterized for total miRNA cargo, in order to investigate if EVs are enriched in biomarkers which could be involved in functional properties.

Starting from the same number of total sEVs, we followed the HTG EdgeSeq NGS procedure (HTG Molecular Diagnostics), described in the Materials and Methods (Section 4.8), which is a semi-automated workflow enabling the simple loading of EV samples without the need for RNA extraction. The bioinformatic analysis was carried out with Reveal software (HTG). The analysis provides information about the quality of the sequencing, including the quality and quantity of total EV-derived RNA, the number of sequences, and genetic variability. All characterized samples demonstrated acceptable quality standards as required by HTG bioinformatic analysis. Then, the HTG software (v. 1.1.4) calculated the correlations among the three technical replicates of the same sample and between all replicates of the two samples (SEC-sEVs and DG-UC-sEVs). In both cases, the correlations were considered to be high (Pearson coefficient > 0.92; Figure 12).

Differential expression analysis between the two sEV preparations was performed by the DESeq2 method, using DG-UC-sEVs as a reference group [27]. Only a few de-regulated probes were found (66): they are represented by red and blue dots in the volcano plot (Figure 13). The red dots are down-regulated miRNAs and the blue dots are up-regulated miRNAs in SEC-sEVs compared with DG-UC-sEVs. Both blue and red dots are miRNAs with an adjusted *p* value < 0.05, meaning that the difference in their expression between the two samples was statistically significant [28]. Instead, grey dots are all the miRNAs for which expression was not found to be different between the two samples. Since very few miRNAs were found to be de-regulated, we can conclude that the two sEV samples are very close to each other. GO analysis of the 66 de-regulated miRNAs was performed (Cellular Component, Biological Process, and Molecular Function) and is reported in the Appendix A section entitled “miRNA GO Analysis” (Appendix A). All the statistically significant categories found for the 66 de-regulated probes were not relevant to our study (meaning that they were not related to the functional properties of MSC-EVs). However, we have observed that, even if not statistically significant, more than half of the de-regulated probes are annotated for categories related to extracellular vesicles, providing further confirmation of the samples’ EV identity.

### 2.4. In Vitro Functional Properties of MSC-EVs

MSC-EVs are reported to be taken up by endothelial cells and to stimulate cell proliferation, differentiation, and migration in vitro [21,22]. We evaluated the functional properties of SEC- and DG-UC-purified MSC-sEVs from 2D and 3D cultures with different cell-based assays (scratch wound healing, proliferation, and angiogenesis assays) using Human Umbilical Vein Endothelial Cells (HUVECs) as the cell model. In the literature, there is no standardization to express the doses of MSC-EVs required for cell treatment. In fact, some studies reported doses expressed as micrograms of MSC-EVs; others as number of particles, but measured with other techniques that are not fully comparable with Flow NanoAnalyzer [21,29,30]. We decided to express the doses of EV treatment as the ratio of the number of sEVs to cells in order to easily translate the same doses into different cell-based assays which require a different number of cells/well.

#### 2.4.1. Scratch Wound Healing Assay Results

The scratch wound healing assay was performed to evaluate the impact of MSC-sEVs on HUVECs’ migration. The protocol of the assay, described in the Materials and Methods (Section 4.9), includes a starvation step to synchronize the cells and halt cell proliferation, thus the gap closure is achieved mainly by cell migration.

Untreated HUVECs (control group) were able to close the gap by 90% after 24 h, and completely after 48 h. After making the scratch, we tested different ratios of MSC-sEVs in order to investigate if the treatment with MSC-sEVs increases the rate of gap closure. Differences in the gap closure rate between untreated and treated cells are visible only after 18 and 24 h. Later differences are not clearly visible, meaning that the gap closure reaches to 100% within the all groups. In both cases (SEC- and DG-UC-sEVs), the addition of sEVs, starting from a 1:1000 cell–EV ratio, enhanced the gap closure rate compared with the control group. The gap closure rate was higher for cells treated with SEC-sEVs than control cells (55% vs. 48% at 24 h with a 1:1000 ratio; 92% vs. 81% at 24 h with a 1:2000 and 1:3000 ratio). Nevertheless, even at the highest dose (ratio of 1:3000), these differences were not statistically significant (*p* value > 0.05). Instead, when cells were treated with DG-UC-derived MSC-sEVs, the difference from the control group was statistically significant (*p* value < 0.05), starting from a lower ratio (1:2000) (96% vs. 81% at 24 h; Figure 14 and Figure 15).

#### 2.4.2. Cell Proliferation Assay Results

To evaluate the impact of MSC-EVs on HUVECs’ proliferation, cells were pretreated with different ratios of DG-UC- and SEC- derived MSC-sEVs for 24 h. Pretreatment of 24 h allows the internalization of EVs by HUVECs. Then, a colorimetric MTT-based assay was used, following the protocol described in Section 4.10. DG-UC-sEVs significantly increased HUVECs’ proliferation, starting with as little as 1000 sEVs per cell (+15% for the 1:1000 and 1:2000 ratios vs. the control group; +28% for the 1:3000 ratio vs. the control group; *p* value < 0.05; Figure 16). SEC-sEVs improved cell proliferation with a statistically significant difference compared with the control group using a ratio of 1:3000 (+19%, *p* value < 0.05; Figure 16).

#### 2.4.3. Tube Formation Assay Results

The tube formation assay was used to evaluate the impact of MSC-EVs on angiogenesis, a phenomenon that includes different processes, such as endothelial cell proliferation, differentiation, and migration. Pre-screened HUVECs, in complete medium seeded on 3D gel, are able to form tube-like structures within 2–4 h and to grow according to a meshed network [31,32]. Since tube formation starts in few hours after cell seeding on Matrigel, we decided to perform a pretreatment of 24 h with MSC-sEVs in order to allow them to be internalized by endothelial cells.

It has been demonstrated that the kinetics of the meshing network structuration of endothelial cells on Matrigel starts with the formation of a multitude of small and unstable meshes. The size of the meshes then progressively increases by fusion of two adjacent meshes, which occurs by the segments’ (tube) regression or disruption [33]. We observed this behavior for both untreated HUVECs and pretreated HUVECs (Figure 17, looking from left to right).

Figure 18 and Figure 19 show representative angiogenesis network parameters, quantified with ImageJ at a time of 10 h.

Our data suggest the formation of more complete networks, with fewer tubes but with bigger meshes in wells with HUVECs pretreated with MSC-sEVs. In untreated wells, clusters of cells that are not organized in tube-like structures are visible even after 20 h from cell seeding (Figure 17). It is challenging to compare these data with previously published reports. In fact, most studies on in vitro pro-angiogenic functionalities of MSC-EVs reported only the increase in some metrics, like nodes or segment length, after treatment with EVs, while neglecting others such as mesh size [22,29,34,35]. However, these data warrant further work to better establish dose ranges and demonstrate the effective internalization of EVs by the cells.

## 3. Discussion

Cell-free therapies based on MSC-EVs are described in the literature as a promising platform for the treatment of different diseases [1,9,16,17]. In this study, we performed an extensive characterization of small EVs isolated from human bone marrow MSC-derived CM, including a study of their functional properties in vitro. We aimed to investigate the impacts of cell culture conditions (i.e., production method) and the purification method on the sEVs’ in vitro functionality. MSC-sEVs were isolated from two sources of CM, a static 2D culture and a suspension 3D culture using a 3 L bioreactor system, suitable for industrial manufacturing, and by two purification methods, SEC and DG-UC. To purify EVs, SEC is a fast and non-laborious method, usually resulting in medium to high EV purity, integrity (>70%), and functionality. In addition, new types of SEC technologies have been optimized recently, which are scalable for industrial manufacturing. On the contrary, DG-UC is considered the gold standard EV purification method, which results in higher EV purity, but it is to be preferred for small studies of proof of concept, because it is not scalable, is time-consuming, and is not suitable for EV manufacturing [36].

First, the purified MSC-sEVs were analyzed for particle size and concentration, and EV purity, integrity, and identity. Consistent with several studies reported in the literature, higher yields of sEVs from 3D culture were obtained compared with the 2D static culture [37,38,39]. Moreover, average particle size was more homogeneous in 3D-derived sEVs, corresponding to small EV subtypes or exosomes (Figure 3b and Figure 5C,D). Finally, sEVs generated using 3D cell culture showed the highest purity indexes, suggesting that the bioreactor-based production is a more manufacturing-friendly method [23,38,40]. As expected, sEVs purified by DG-UC had a higher purity index compared with SEC-derived sEVs for both 2D and 3D CMs. Co-purified proteins could not be fully separated from EVs by SEC. Thus, further comparisons with new technologies (e.g., superSEC; Cytiva) or other purification techniques (e.g., anion-exchange chromatography) will be required [41,42,43]. Despite the different purity indexes, both sEV types resulted in similar integrity and expression of canonical sEV markers, confirming the presence of intact vesicles with typical exosomal phenotypes. The similarity between the two sEV-purified samples was also confirmed by analyzing their protein and miRNA cargo (limited to 2D-derived samples). Bioinformatic analysis of the proteome and miRNA cargo did not show significant differences in the two sEV preparations, since the correlation coefficients were high and few probes were de-regulated between the samples. The few differences between SEC-sEVs and DG-sEVs could be explained by the more heterogenous nature of SEC-derived EVs, usually co-purified with the associated biomolecules. However, both sEV groups are enriched in biomarkers annotated for EV-related categories and contain some proteins which could exert functional effects in vitro.

Then, we investigated the in vitro functional properties of MSC-sEVs, establishing complementary cell-based assays which allow to see the impact of EV treatment on endothelial cell proliferation, migration, and differentiation. We found that MSC-sEVs from both 2D and 3D sources were able to increase endothelial cell capabilities like proliferation and migration. In fact, HUVECs pretreated or treated with MSC-sEVs at different doses showed significant different behaviors compared with untreated HUVECs, proving the use of simple cell-based assays like quantification of cell growth through MTT labeling or more complex models like tube formation on 3D gel. On the basis of the previously reported data on EVs’ identity, integrity and purity, we can speculate that these functional effects are mediated mainly by small EVs (exosomes) present in the MSC secretome. Lower doses of DG-UC-sEVs were needed to obtain functional effects that were statistically significant compared with the control group. The different doses needed for SEC- and DG-UC-sEV samples could be explained by the presence of fewer contaminants in the purer samples or by a population of heterogeneous EVs in the case of SEC-derived samples, which could have different and contrasting effects.

Moreover, in the body, most MSCs exist in a hypoxic environment: 1–9% O_2_ in bone marrow, 5–9% O_2_ in adipose tissue, and 1–6% O_2_ in umbilical cord blood [44,45]. On the contrary, MSCs for in vitro experiments are usually cultured in 21% O_2_. It has been reported that an in vitro hypoxic environment improves the proliferation and migration of MSCs during the expansion process, as well as promoting their differentiation ability, compared with MSCs cultured under normoxia [46]. Similarly, hypoxia can also influence the secretion of cytokines, growth factors, and EVs [47]. In particular, MSC-EVs generated under hypoxia seem to have improved angiogenic effects and enhanced cytoprotective and anti-inflammatory properties [48]. For this reason, future studies are warranted to evaluate if MSC-EVs released under hypoxic conditions have improved functional properties in vitro and in vivo, compared with MSC-EVs generated under normoxia.

## 4. Materials and Methods

### 4.1. MSC Culture

Both 2D- and 3D-based cultures of MSCs were performed and the CM was collected, respectively, after 13 and 11 days. For 2D culture, human bone marrow mesenchymal stem cells (Lonza, Basel, Switzerland) were thawed and plated in T-175 cm^2^ flasks in Lonza MSC medium without serum for 3 days, replacing the media after 24 h of plating. After 3 days, the cells were passaged into T-275 cm^2^ flasks and monitored daily. Once the culture reached ~80% confluence, cells were harvested and passaged into CF10 flasks. The cell culture was carried out for 5 days to allow the cells to become 100% confluent and another 2 days for EV production.

At the end, CM was harvested and pre-cleared by differential centrifugation at 300× *g* for 10 min to remove any cells in the suspension, and at 1200× *g* for 20 min and 10,000× *g* for 30 min at 4 °C to remove any cell debris and large EVs in the suspension. Pre-cleared CM was frozen at −80 °C.

For 3D culture, a 3 L microcarrier-based bioreactor was used. MSCs were thawed and expanded in T225 flasks. After 3 days of culture, MSCs were harvested, counted, and inoculated into the bioreactor. After 8 days, microcarriers were digested, and the CM was harvested, pre-cleared by centrifugation as stated before for 2D culture, aliquoted, and frozen at −80 °C.

### 4.2. Density Gradient Ultracentrifugation (DG-UC)

For this, 200 mL of pre-cleared MSC-CM was thawed and concentrated up to 10 mL by ultrafiltration with an Amicon^®^ Stirred Cell with Ultracel 100 kDa Ultrafiltration Discs (Merck Millipore, Burlington, MA, USA).

Since the SEC protocol required the loading of 10 mL of concentrated CM, while DG-UC needed only 5 mL, two DG-UC purifications were pooled into one final EV sample. Small EVs were purified using an OptiPrep™ (Merck Millipore, Burlington, MA, USA) cushion density flotation. The iodixanol gradient was prepared by floating 3 mL of a 10% *w*/*v* iodixanol solution containing NaCl (150 mM) and Tris HCl (25 mM, pH 7.4) over 3 mL of a 55% *w*/*v* iodixanol solution. Concentrated CM (5 mL) was floated on top of the iodixanol cushion and ultracentrifuged using a Fiberlite F65L rotor (Thermo Fisher Scientific, Cleveland, OH, USA) for 3.5 h at 100,000× *g*. Eleven fractions (1 mL each) were collected from the top of the gradient and kept at −80 °C.

### 4.3. Size Exclusion Chromatography (SEC)

For each condition, 200 mL of pre-cleared CM was concentrated by ultrafiltration with an Amicon^®^ Stirred Cell with Ultracel 100 kDa Ultrafiltration Discs (Merck Millipore, Burlington, MA, USA) to a final volume of 10 mL. Concentrated samples were purified by SEC (qEV10 35 nm, Izon Science, Christchurch, New Zealand). Columns were pre-equilibrated with 1X 0.22 µm filtered PBS, 10 mL of concentrated CM was loaded in the column, and then 1X 0.22 µm filtered PBS was used as the mobile phase. The eluate was immediately collected in 60 fractions of 1 mL each. All of the fractions were characterized by a micro BCAassay (Thermo Fisher Scientific, Cleveland, OH, USA), EV particle concentration measured with a Flow NanoAnalyzer (nanoFCM Inc., Nottingham, UK), and by a sandwich ELISA assay targeting a mix of three tetraspanins CD9, CD63, and CD81 (Exbio, Praha, Czech Republic). Positive fractions for the tetraspanins, but with a low protein concentration, were pooled and concentrated to 0.5 mL by 100 kDa ultrafiltration (Amicon^®^ Ultra-15 Centrifugal Filter Unit, Merck Millipore, Burlington, MA, USA). Purified sEVs were aliquoted and stored at −80 °C.

### 4.4. Transmission Electron Microscopy

Transmission Electron Microscopy (TEM) was used to verify EVs’ morphology and integrity. Each sEV sample (3 μL, corresponding to 0.12–0.16 µg) was loaded for 2 min onto a 300 mesh formvar-coated copper grid. After blotting the excess, the grid was negatively stained with 2% aqueous ammonium molybdate for 30 s and analyzed using a Thermo Fisher Scientific Tecnai G2 Spirit transmission electron microscope operating at 120 kV equipped with a EMSIS Veleta 2048X2048 CCD camera (Thermo Fisher Scientific, Cleveland, OH, USA).

### 4.5. Flow NanoAnalyzer Analysis

EV samples were analyzed using a Flow NanoAnalyzer (nanoFCM Inc., Nottingham, UK) to measure the EV size and particle concentration. Bar charts are representative of three independent experiments. The instrument was aligned and calibrated with size and concentration standard beads (nanoFCM Inc., Nottingham, UK), which allowed for the characterization of small EVs between 40 nm and 200 nm. The samples and blank (1X PBS) were read at a constant pressure of 1 kPa for 2 min and at a maximum event rate of 12,000 events/min. Between samples, the instrument was cleaned with 1X cleaning solution (nanoFCM Inc., Nottingham, UK) and the capillary was rinsed with HPLC-grade water.

In addition, sEV samples were stained with a CellTrace CFSE Cell Proliferation kit (Thermo Fisher Scientific, Cleveland, OH, USA) to evaluate EV integrity. The same number of particles was incubated with CFSE 10 µM for 1.5 h at 37 °C under shaking. Then, the excess dye was removed with a SEC qEV original column (Izon Science, Christchurch, New Zealand).

To confirm EVs’ identity, samples were stained with a mix of three tetraspanin antibodies (CD9, CD63, and CD81; Exbio, Praha, Czech Republic) conjugated with a fluorophore. The same number of particles was incubated with antibodies for 1 h at 37 °C under shaking. Then, the excess dye was removed with centrifugal filters (Pall Corporation, Porth Washington, NY, USA). Fluorescent samples were analyzed using a Flow NanoAnalyzer in order to assess the percentage of positive events.

### 4.6. Western Blotting

A Western blot assay of purified sEVs, from 2D and 3D cultures, was run to check the expression of canonical EV markers and negative EV markers. sEV samples were prepared by mixing 2 × 109 total particles with Laemmli sample buffer (4×; Bio-Rad, Hercules, CA, USA) containing beta-mercaptoethanol and incubated at 95 °C for 10 min. The same procedure was followed for cell lysates mixing a total of 30 µg of proteins. Samples were loaded on precast 4–20% Mini-PROTEAN^®^ TGX Gel, 12-well, at 20 µL per well (Bio-Rad, Hercules, CA, USA), then the proteins were transferred onto a 0.2 µm nitrocellulose membrane using the Trans-Blot Turbo Transfer System with the Trans-Blot Turbo Transfer Pack (Bio-Rad, Hercules, CA, USA). Western blotting was performed using EveryBlot Blocking Buffer (Bio-Rad, Hercules, CA, USA) for the blocking step and to dilute the primary antibodies and the secondary antibodies.

Primary antibodies against CD81 (BD Biosciences, San Jose, CA, USA, 1:500, Cat. No. 555675), Alix (Santa Cruz, Dallas, TX, USA, 1:500, Cat. No. sc-271975) and calnexin (Santa Cruz, Dallas, TX, USA, 1:200, Cat. No. 11397) were used.

### 4.7. Mass Spectrometry

To measure the total protein content, sEV samples were quantified using a microBCA assay (Thermo Fisher Scientific, Cleveland, OH, USA). The same amount of protein per sample was used for acetone-based precipitation. Samples were precipitated with chilled acetone (Merck Millipore, Burlington, MA, USA), mixed by vortexing, and incubated at −20 °C overnight.

The precipitated protein pellets were dried and solubilized with urea 8 M Tris-HCl 100 mM pH 8.5 (Merck Millipore, Burlington, MA, USA). Protein quantification was repeated with the microBCA assay.

For the protein digestion step, a minimum of 10 µg of proteins were digested with Trypsin Gold Mass Spectrometry Grade (Promega, Madison, WI, USA) at 37 °C overnight.

The resultant peptides were resuspended in 0.1% trifluoroacetic acid water at a final concentration of 1 µg/µL. A total of 3 µL (equal to 3 µg of starting material) was injected into the Nano LC-MS apparatus (Q Exactive HF X-UHPLC Nano; Thermo Fisher Scientific, Cleveland, OH, USA).

Peptide separation was carried out at 35 °C using a PepMap RSLC C18 column, 75 µm × 15 cm, 2 μm, 100 Å (Thermo Fisher Scientific, Cleveland, OH, USA), at a flow rate of 300 nL/min. The mobile phases A and B used for the analysis were 0.1% formic acid in water and 0.1% formic acid in acetonitrile, respectively.

Protein identification was performed by Proteome Discoverer 2.5 (Thermo Fisher Scientific, Cleveland, OH, USA). The peptide spectra were matched against the Homo sapiens database downloaded from Uniprot (TaxId: 9606). The analysis was based on at least one unique peptide with a minimum length of seven amino acids and a false discovery rate (FDR) of 0.01. The default peak-picking settings were used to process the raw MS files in MaxQuant (version 1.6.1.0) and its integrated search engine, Andromeda [26,49]. The proteins’ relative quantification and calculation of statistical significance were carried out using a two-tailed Student’s *t*-test and error correction (*p* value < 0.05) with the method of Benjamini–Hochberg. Moreover, a volcano plot showing a summary distribution of differentially expressed proteins was designed with Perseus software (version 1.6.1.1) [50]. The obtained proteins were compared with the Exocarta database to check for their presence in the top 100 proteins that are often identified in exosomes [51]. Moreover, by means of the Uniprot database, proteins belonging to the GOs extracellular exosome [GO:0070062] and extracellular region [GO:0005576] were identified.

### 4.8. miRNA Whole Transcriptome Assay (WTA)

The miRNA Whole Transcriptome Assay (WTA) is a next-generation sequencing (NGS)-based application that measures the expression of 2083 human microRNAs (miRNAs) directly from EVs, without the RNA extraction step. The preparation of sequencing libraries was carried out with a semi-automated system (HTG molecular EdgeSeq; HTG Molecular Diagnostics, Tucson, AZ, USA). Small EV samples (three technical replicates for each type of purification method) were lysed at 95 °C with a lysis buffer (HTG Molecular Diagnostics, Tucson, AZ, USA) and then loaded on sample plate (HTG Molecular Diagnostics, Tucson, AZ, USA) for the quantitative nuclease protection assay. Then, sequencing adapters were added to the probes via PCR. The resulting PCR products were cleaned and quantified by qPCR. Then, the normalized pool of libraries was run on a MiSeq^®^ sequencer (Illumina, San Diego, CA, USA).

Data analysis was carried out using Reveal software v. 2.0 (HTG Molecular Diagnostics, Tucson, AZ, USA), which performed differential expression analysis with the DESeq2 package (version 1.30.1) available from Bioconductor [52].

GO analysis was carried out with the miRNA Enrichment Analysis and Annotation Tool (miEAA) [53].

### 4.9. Scratch Wound Healing Assay

The scratch wound healing assay was used to evaluate the impact of MSC-sEVs from 2D and 3D cultures on cell migration in vitro. Human umbilical vein endothelial cells (HUVECs; Lonza, Basel, Switzerland) were seeded into a 24-well plate in EGM™ Medium (Lonza, Basel, Switzerland). After 24 h, cells were starved in order to synchronize them and inhibit cell proliferation. When the cells were 85–90% confluent, a scratch was performed in the center of the cell monolayer. The cell debris was washed with PBS. Then, cells were treated with a medium containing MSC-sEVs (doses are expressed as the cell–EV ratio). Each condition was performed in three technical replicates. Pictures were taken under a microscope at 0, 18, and 24 h after creating the scratch. The scratch area at different timepoints was measured using ImageJ software v. 1.54d. The results are expressed as the percentage rate of gap closure at times of 18 h or 24 h, compared with Time 0. The percentage rate was calculated using the formula (1 − Ax/A0)%, where A0 and Ax represent the empty scratch area at 0 h and x hours (x = 18 h or 24 h).

### 4.10. MTT Proliferation Assay

HUVECs (Lonza, Basel, Switzerland) were seeded into 96-well plates in EGM medium (Lonza, Basel, Switzerland) and cultured overnight. Cells were starved for at least 16 h in order to synchronize them. Then, cells were incubated with different amounts of sEVs from 2D and 3D cultures (expressed as the cell–EV ratio) for 24 h. Each condition was performed in six replicates. Cell proliferation was analyzed using the Cell Proliferation Kit I (MTT; Roche, Basel, Switzerland). For this, 10 µL of the MTT solution was added to the cells and incubated at 37 °C for 4 h. A solubilization buffer was added to dissolve the purple crystals overnight at 37 °C. Then, the absorbance was read at 570 nm (reference wavelength > 650 nm) using a microplate reader (CLARIOStar Plus, BMG Labtech, Ortenberg, Germany). The background was removed by subtracting the absorbance recorded from a well without cells.

### 4.11. Tube Formation Assay

HUVECs pre-screened for angiogenesis (Lonza, Basel, Switzerland) were cultured using EGM™-2 Medium, containing VEGF (Lonza, Basel, Switzerland). Cells were first seeded into 6-well plates at a density of 6 × 105 cells/well. After 24 h, they were starved for at least 16 h to ensure that all cells were synchronized at the same cell cycle stage. Then, the cells were pretreated with 2D and 3D MSC SEC-derived sEVs or DG-derived sEVs (1:1000 cell–EV ratio) for 24 h.

The day after, cells were detached and plated at a density of 3 × 10^4^ cells/well in triplicate for each condition in 96-well plates coated with 75 µL of Matrigel Basement Membrane Matrix (Corning, Corning, NY, USA).

Pictures were taken under a microscope after 2, 4, 10, and 20 h at 4× magnification. Images were analyzed using ImageJ software v. 1.54d with the Angiogenesis Analyzer plugin for quantification of tube networks [31].

### 4.12. Statistical Analysis

Purification of MSC-derived sEVs from 2D and 3D cell culture was carried out in triplicate for each isolation method.

Prism Version 10.00 for Windows (GraphPad Software, La Jolla, CA, USA) was used for statistical analysis. Results of the analytical methods are reported as the mean ± standard deviation (SD) of at least two technical replicates. Error bars show the SD of the mean.

A two-tailed Student’s *t*-test was applied to evaluate the statistical difference between two conditions. Statistical comparisons that output *p* values < 0.05 were considered significant.

## Figures and Tables

**Figure 1 ijms-26-10602-f001:**
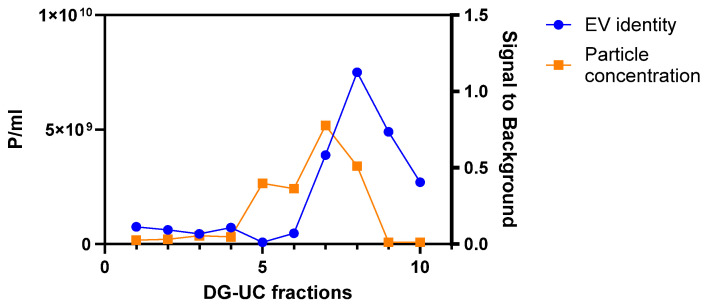
Example of an overlay of particle concentration (in orange) and positivity signal for three tetraspanins (in blue) of 10 fractions obtained with one DG-UC purification run of MSC CM derived from 2D culture.

**Figure 2 ijms-26-10602-f002:**
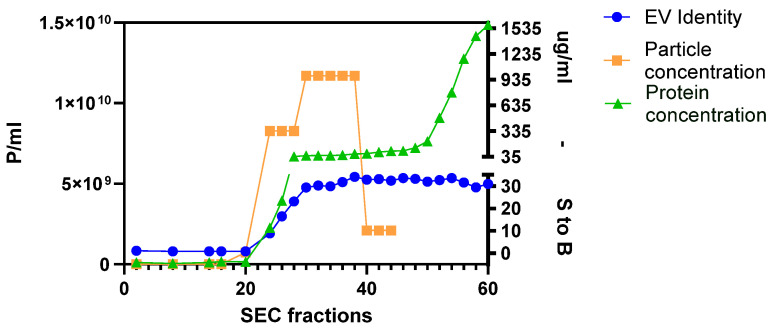
Example of one SEC elution profile of MSC CM derived from 2D culture. The graph shows the overlay of particle concentration (in orange), protein concentration (in green), and positivity for three tetraspanins (in blue; S to B: signal to background) of 60 fractions. For this experiment, fractions from 24 to 38 were pooled and concentrated to obtain the final sEV sample.

**Figure 3 ijms-26-10602-f003:**
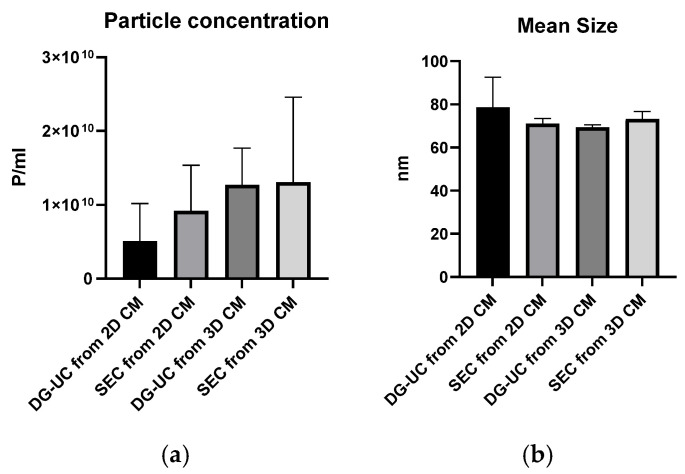
Particle concentration (**a**) and particle mean size (**b**) of sEV samples purified by SEC and DG-UC from 2D and 3D cultures measured by a Flow NanoAnalyzer. Bar charts are representative of three independent purifications.

**Figure 4 ijms-26-10602-f004:**
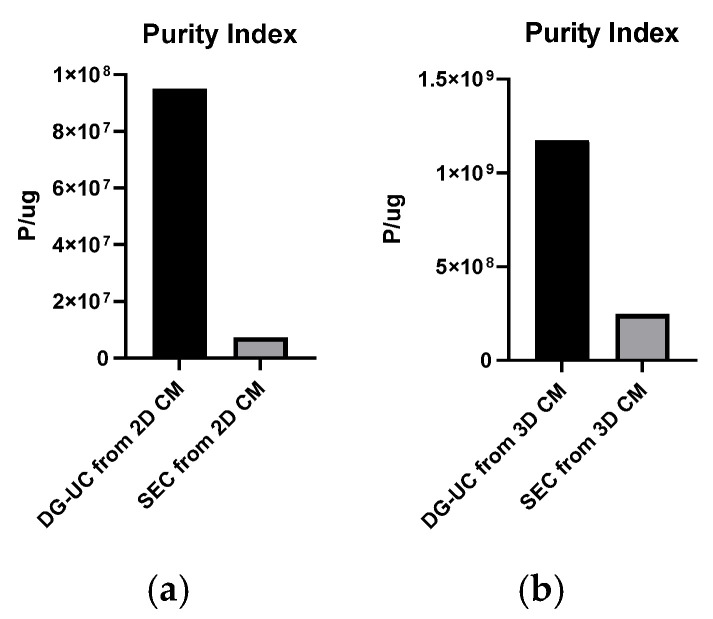
Purity indexes of sEV samples obtained from 2D (**a**) and 3D cultures (**b**).

**Figure 5 ijms-26-10602-f005:**
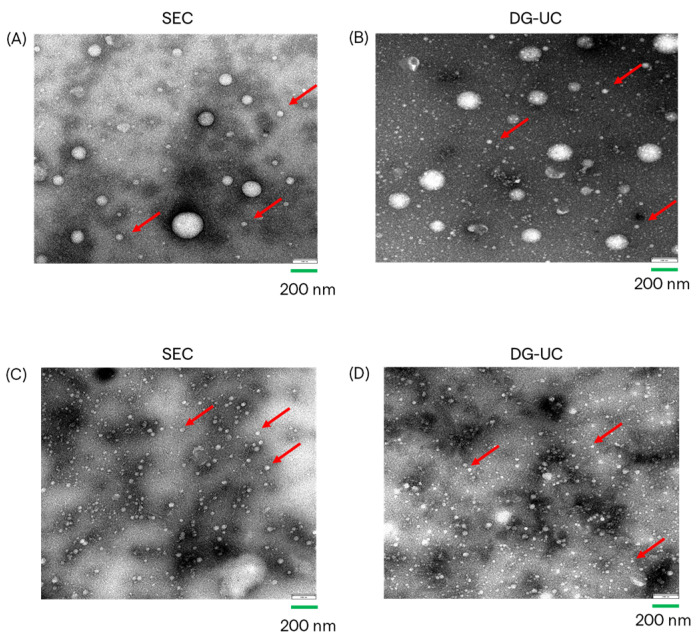
TEM images of MSC-EVs from 2D CM, purified by SEC (**A**) and by DG-UC (**B**); TEM images of MSC-EVs from 3D CM, purified by SEC (**C**) and by DG-UC (**D**). Red arrows indicate small EVs.

**Figure 6 ijms-26-10602-f006:**
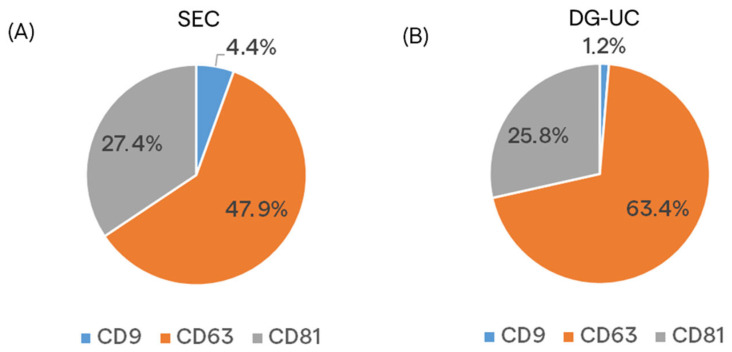
Pie charts representing the percentage of staining of the single tetraspanins in sEV samples purified by SEC (**A**) and DG-UC (**B**) from MSC 3D culture.

**Figure 7 ijms-26-10602-f007:**
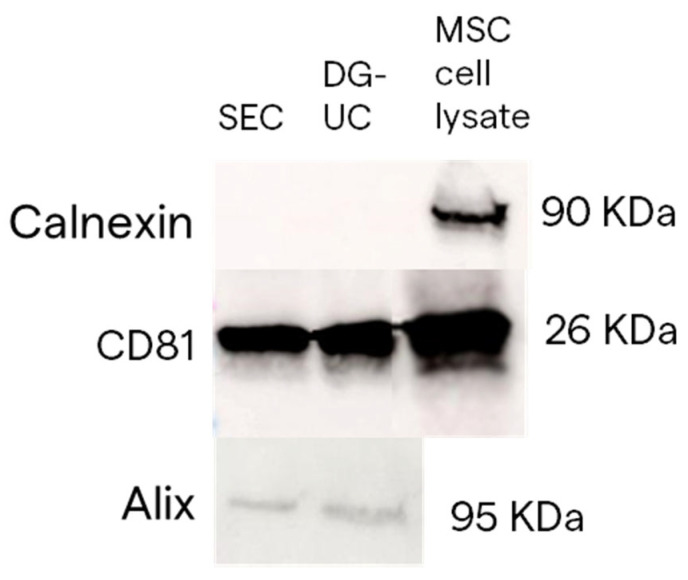
Example of blots representing CD81 and Alix protein expression in sEV samples obtained by SEC and DG-UC from 2D MSC culture. Calnexin protein expression was used as an EV-negative control.

**Figure 8 ijms-26-10602-f008:**
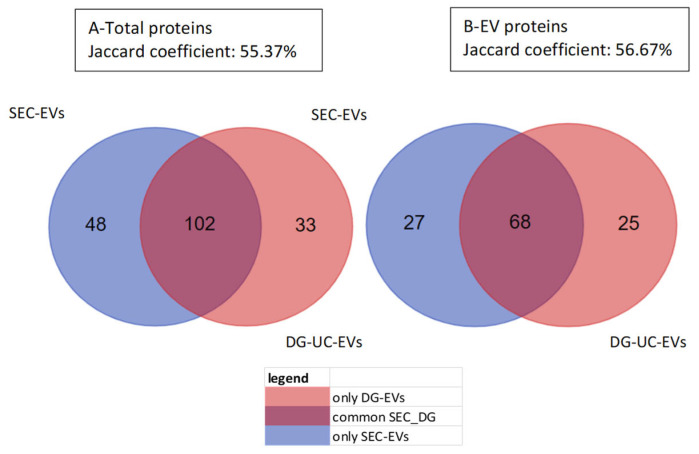
Venn charts of total proteins (**A**) and only EV-related proteins (**B**) detected in SEC-sEVs, in DG-UC-sEVs, and in common between the two groups. Samples were run in duplicate and analysis was carried out considering 2 peptides.

**Figure 9 ijms-26-10602-f009:**
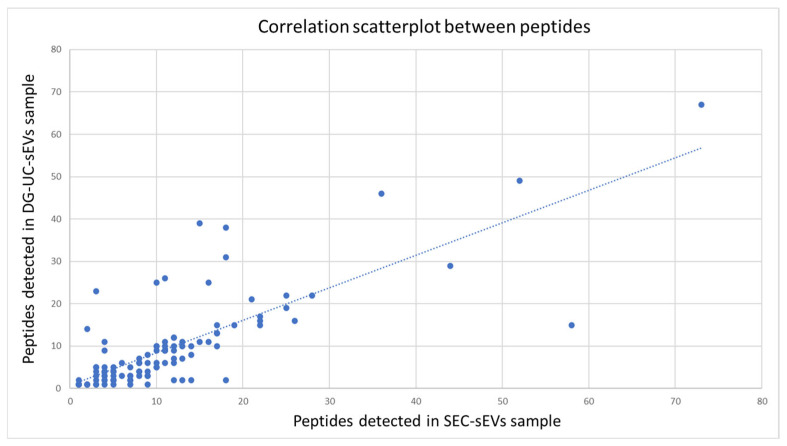
Correlation analysis of the subset of common proteins detected in the two sEV samples: scatterplot between the groups of peptides detected in SEC-sEVs (x-axis) and in DG-UC-sEVs (y-axis). Most of the points (peptides) are clustered around a straight line, indicating a medium–high correlation (correlation coefficient: 0.76).

**Figure 10 ijms-26-10602-f010:**
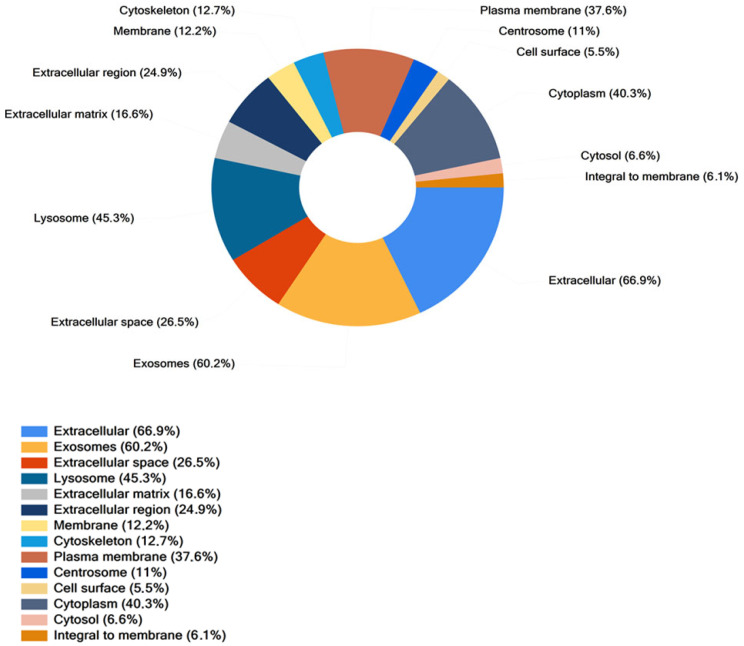
Cellular Component analysis of proteins found in common between SEC-sEVs and DG-UC-sEVs. Samples have been run in duplicate.

**Figure 11 ijms-26-10602-f011:**
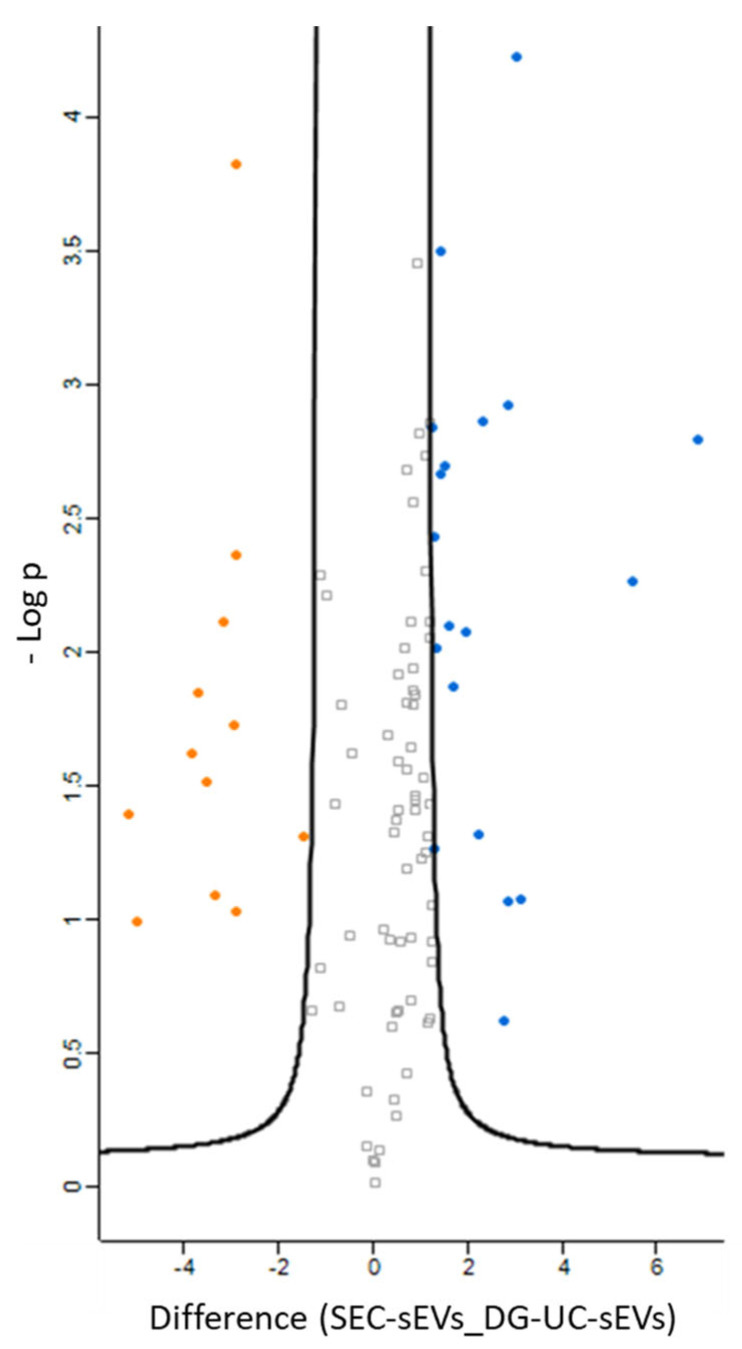
Volcano plot showing significantly differentially expressed proteins in SEC-sEVs (blue dots) and DG-UC-sEVs (orange dots). Grey dots are proteins that were not significantly differentially expressed. Samples were run duplicate.

**Figure 12 ijms-26-10602-f012:**
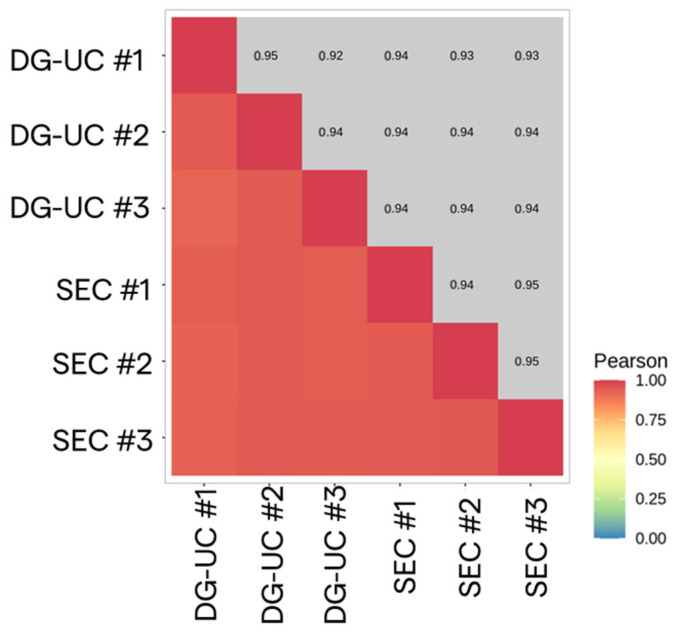
Correlation heatmap between SEC-sEV (n = 3 replicates) and DG-UC-sEV (n = 3 replicates) samples.

**Figure 13 ijms-26-10602-f013:**
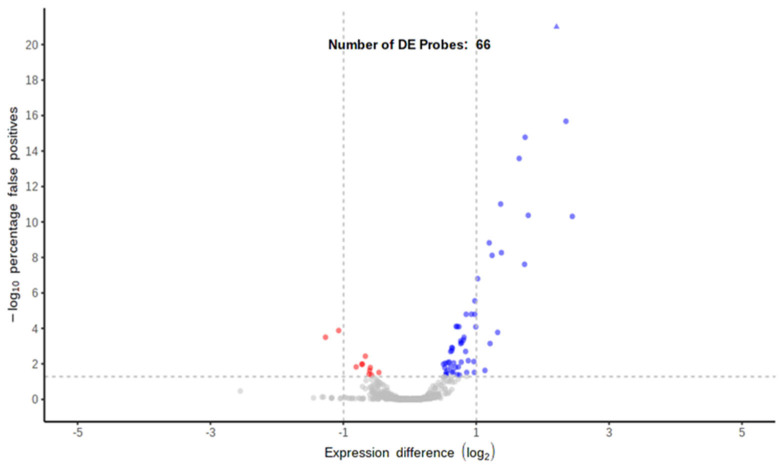
Volcano plot representing differentially expressed probes using DG-UC-sEVs as the reference group (red dots are down-regulated miRNAs and blue dots are up-regulated miRNAs; grey dots are miRNAs not differentially expressed between the two samples).

**Figure 14 ijms-26-10602-f014:**
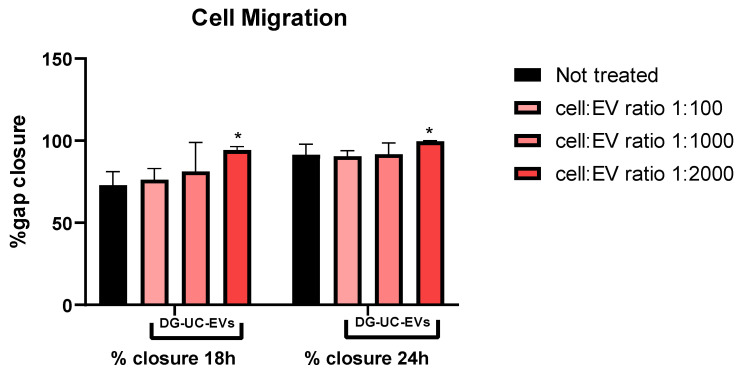
Example of the rate of wound closure calculated for HUVECs not treated and treated with DG-UC-sEVs from 2D culture. Bar chart is representative of three technical replicates. * indicates a statistically significant difference (*p* value < 0.05).

**Figure 15 ijms-26-10602-f015:**
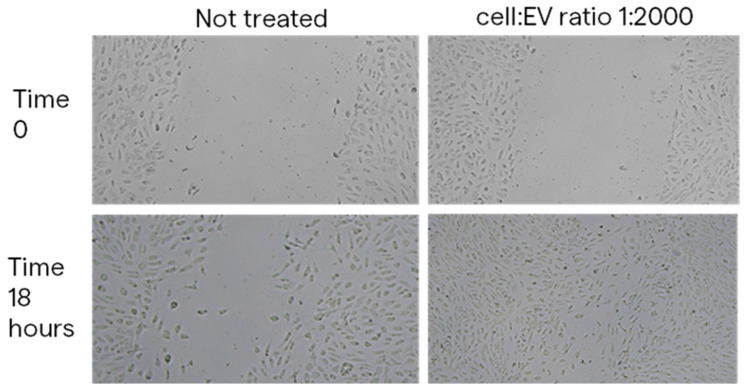
Images at 4× magnification of the wound space at Time 0 and 18 h without treatment and with treatment with DG-UC-sEVs from 2D culture at a cell–EV ratio of 1:2000.

**Figure 16 ijms-26-10602-f016:**
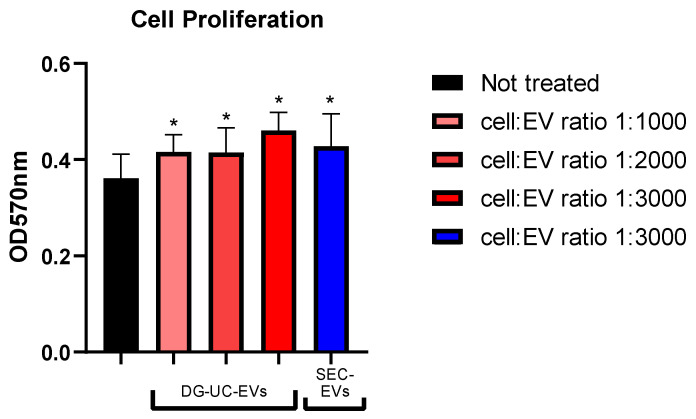
MTT-based colorimetric assay to quantify HUVEC proliferation, with or without a pretreatment with MSC-sEVs from 3D culture. Bar chart is representative of six technical replicates. * indicates a statistically significant difference (*p* value < 0.05).

**Figure 17 ijms-26-10602-f017:**
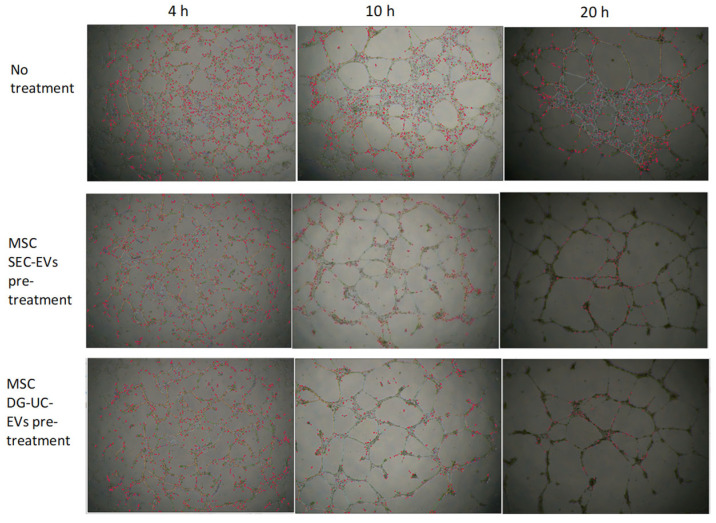
HUVEC network analysis of pictures taken at 4, 10, and 20 h, using microscope’s magnification 4×, after cell seeding on Matrigel. The pretreatment with MSC-sEVs, both SEC-derived and DG-UC-derived sEVs from 3D culture, seems to have an impact on HUVEC tube formation. In fact, it resulted, for all analyzed timepoints, in a significant decrease in most of the parameters quantified by Angiogenesis Analyzer for ImageJ (extremities, nodes, junctions, master junctions, segments, master segments, branches, meshes, segment length, branch length), except for “mesh index” and “mean mesh size”, which, conversely, significantly increased (*p* value < 0.05). Mesh index, which is the mean distance separating two master junctions, represents a further indication of mesh size.

**Figure 18 ijms-26-10602-f018:**
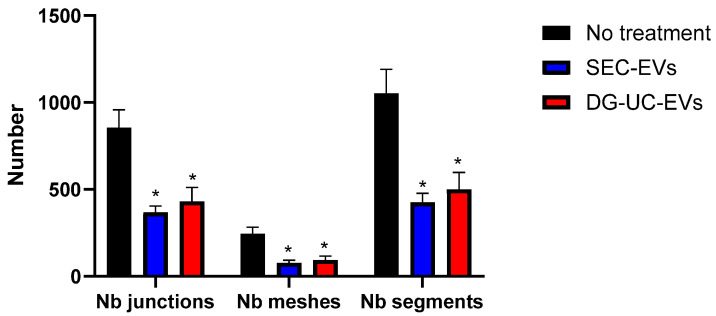
Quantification of representative angiogenesis network parameters at 10 h in HUVECs without and with pretreatment with MSC-sEVs from 3D culture. Bar chart is representative of three technical replicates. * indicates a statistically significant difference (*p* value < 0.05).

**Figure 19 ijms-26-10602-f019:**
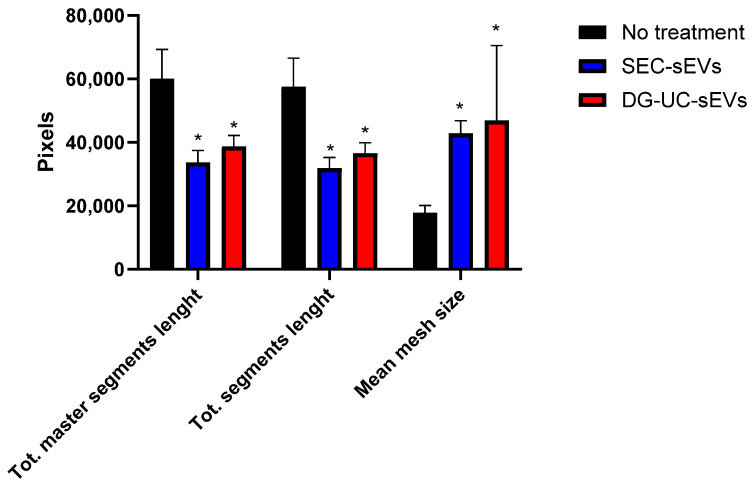
Quantification of representative network parameters at 10 h in HUVECs without and with pretreatment with MSC-sEVs from 3D culture. Bar chart is representative of three technical replicates. * indicates a statistically significant difference (*p* value < 0.05).

## Data Availability

The original contributions presented in this study are included in the article/Appendix A. Further inquiries can be directed to the corresponding author(s).

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
