# Peer review of "Comprehensive Characterization and In Vitro Functionality Study of Small Extracellular Vesicles Isolated by Different Purification Methods from Mesenchymal Stem Cell Cultures"

_ijms, 2025, doi:10.3390/ijms262110602_

Round 1
Reviewer 1 Report
Comments and Suggestions for Authors
Dear authors,
I was pleased to have the opportunity to read and review an interesting, up-to-date article, „ Investigation of in vitro regenerative effects of small Extracellular Vesicles purified with different methods from Mesenchymal stem cells 2D and 3D culture,“ that involved an enormous amount of work, material collection, and analysis. I really appreciate the complexity of the view on EVs from isolation, characterization to studying the effect. The article is written clearly with a logical sequence and continuity.
I just have only few comments and questions.
Material and Methods - MSC culture and collection of CM - How long (in days ) have you been collecting CM for 2D- and 3D-based culture?
Results - Figure 1: Is it from 2D or 3D culture?
- Which concrete fractions of EVs isolated by SEC (2D, 3D) (with high particle concentration (orange line) and positive for the three tetraspanins CD9-CD63-CD81 (blue line), but with a low protein concentration (green line)) did you identify (fractions 20-40, or 22-41?) and use later?
The article will certainly be helpful for people working with EVs isolated from MSCs. Therefore, with minimal modifications, I recommend it for publication.
Author Response
Please see the attachment:
1. Summary
Thank you very much for taking the time to review this manuscript. Please find the detailed responses below and the corresponding revisions/corrections highlighted/in track changes in the re-submitted files.
2. Questions for General Evaluation
Reviewer’s Evaluation
Response and Revisions
Does the introduction provide sufficient background and include all relevant references?
Yes
Are all the cited references relevant to the research?
Yes
Is the research design appropriate?
Yes
Are the methods adequately described?
Yes
Are the results clearly presented?
Yes
Are the conclusions supported by the results?
Yes
3. Point-by-point response to Comments and Suggestions for Authors
Comments 1: Material and Methods - MSC culture and collection of CM - How long (in days ) have you been collecting CM for 2D- and 3D-based culture?
Response 1: Thank you for pointing this out. CM was collected after 13 days for 2D culture and 11 days for 3D culture. This clarification has been added to Material and Methods section 4.1. MSC culture (Pag. 16).
Comments 2: Results - Figure 1: Is it from 2D or 3D culture?
Response 2: It is an example of DG-UC purification from 2D-derived CM. This information has been added in the figure legend (Pag. 3).
Comments 3: Which concrete fractions of EVs isolated by SEC (2D, 3D) (with high particle concentration (orange line) and positive for the three tetraspanins CD9-CD63-CD81 (blue line), but with a low protein concentration (green line)) did you identify (fractions 20-40, or 22-41?) and use later?
Response 3: We took fractions starting from fr. 24 up to fr. 38. This information has been added in the figure legend (Pag. 4).
Reviewer 2 Report
Comments and Suggestions for Authors
In the manuscript entitled “Investigation of in vitro regenerative effects of small Extracellular Vesicles purified with different methods from Mesenchymal stem cells 2D and 3D culture” authors aimed to characterize 2D and 3D culture MSC derived EVs isolated by SEC and DG-UC, and evaluate their biological effects on HUVECs. Based on the results obtained, authors concluded that both EV samples have functional effects on endothelial cells compared to the untreated control, most notably the EVs isolated by DG-UC. The paper is very interesting, with a lot of technics used, and can be accepted after revision of some minor/major issues.
- Title
The title could be revised to include characterization of EVs as well as their methods of isolation (SEC and DG-UC) since largest part of article is about EVs characterization and comparison between these two methods (preferable instead of 2D/3D culture).
- Results
Lines 81-83 Sentence: Since SEC protocol required… is more appropriate to Materials and methods section
Lines 89-90 Authors should include in every figure legend whether the results are of EVs derived from MSC 2D or 3D culture.
Also, for every technique it must be stated in text whether the results are from 2D or 3D culture derived EVs.
MAJOR ISSUE – why are some experiments performed only on 2D culture and other only on 3D culture derived EVs? There are also some experiments where EVs from both cultures are compared. Authors must address this issue thoroughly, explain it, and make changes to manuscript text accordingly. Moreover, the authors stated in Discussion section that EVs derived from MSC 3D culture showed highest purity, it would be logical that after such characterization focus stays on these EVs.
Lines 113-116 Please specify range of pooled fractions
Line 123 Purification runs were done in triplicate – include this statement in figure 2 also.
Line 135 – swap upper case letters A and B to small letter case.
Lines 136-137 representative of three independent experiments – should be also stated in materials and methods section
Line 155 MVs – first time introduced abbreviation, include explanation for it.
Lines 158-159 – swap upper case letters A and B to small letter case.
Lines 177-184 – Specify here and in materials and methods section for western blot, which EVs sample is analyzed – from 2D or 3D culture.
Lines 202, 205, 206 etc. – uniform sample names throughout manuscript and in all figures/figure legends – i.e. SEC-sEVs or SEC-EVs.
Line 208 – small proteins should be revised to either proteins or associated proteins, because larger proteins could also be contaminants, either by being soluble or associated as part of “soft” corona of EVs.
Line 242 – although authors stated it as limitations, they should explain why transcriptomic analysis was performed only with MSC-derived EVs purified from 2D CM.
Line 282 – which MSC-EVs were used in these experiments? From 2D or 3D culture? Please specify here and in materials and methods section
Figures 15 and 17 – please change in figure DG-UC to DG-UC-EVs and SEC to SEC-EVs
Figures 19 and 20 - DG-UC-EVs and SEC-EVs instead of DG-UC-sEVs and SEC-sEVs
- Supplementary
Please revise “compared to the second sample” for clarification on what is second sample – “Table S1 reports the 31 proteins that are up-regulated in SEC- and DG-UC-EVs, respectively compared to the second sample”.
- Discussion
Lines 397-398 – include this reference too since it’s the first time anion exchange chromatography was used for purification of EVs [Kosanović M, Milutinović B, Goč S, Mitić N, Janković M. Ion-exchange chromatography purification of extracellular vesicles. Biotechniques. 2017 Aug 1;63(2):65-71. doi: 10.2144/000114575. PMID: 28803541.]
Authors could discuss about some differences in miRNAs between SEC-EVs and DG-UC-EVs, i.e. presence of other EVs in SEC-EVs sample which is more heterogeneous…
Lines 409-410 – in results/materials and methods sections it is not specified which MSC-EVs were used, either from 2D or 3D culture, and figures results are only from one set of experiments 2D or 3D?).
Lines 416-417 – “As a consequence of the higher purity of DG-UC-EVs”, is greater effect of DG-UC-EVs influenced by higher purity of EVs in sample in terms that there are less contaminants which may interfere or that greater heterogeneity of SEC-EVs, i.e. larger vesicles might also influence functional effects? – Very interesting topic that authors could also discuss in a sentence or two.
- Materials and methods
Maybe in future experiments it would be more appropriate to centrifuge obtained CM at 300 x g before freezing in order to avoid possible intracellular vesicle contamination if some cells deteriorate during freeze-thaw cycle/cycles.
Line 441, 459 etc – uniform conditioned medium (CM) – when abbreviation is introduced first time, it should be used throughout the text
Author Response
|
Comment 1: The title could be revised to include characterization of EVs as well as their methods of isolation (SEC and DG-UC) since largest part of article is about EVs characterization and comparison between these two methods (preferable instead of 2D/3D culture). |
|
Response 1: Thank you for pointing this out. We agree with this comment. Title has been modified: “Comprehensive characterization and in vitro functionality study of small Extracellular Vesicles isolated with different purification methods from Mesenchymal Stem Cell cultures”. |
|
Comment 2: Results. Lines 81-83 Sentence: Since SEC protocol required… is more appropriate to Materials and methods section. |
|
Response 2: Agree. We have, accordingly, moved this sentence to Material and Methods section (Pag. 16). Comment 3: Lines 89-90 Authors should include in every figure legend whether the results are of EVs derived from MSC 2D or 3D culture. Response 3: Information has been added in all the figure legends. Comment 4: MAJOR ISSUE – why are some experiments performed only on 2D culture and other only on 3D culture derived EVs? There are also some experiments where EVs from both cultures are compared. Authors must address this issue thoroughly, explain it, and make changes to manuscript text accordingly. Moreover, the authors stated in Discussion section that EVs derived from MSC 3D culture showed highest purity, it would be logical that after such characterization focus stays on these EVs. Response 4: EV purification has been done with two methods, from both the sources of CM culture. The main characterization panel, including the cell-based assays, has been performed for all the samples (SEC/DG-UC for 2D/3D culture), except for single tetraspanins stainings (fig. 6), which was an additional information beside CD9 expression measured by ELISA assay, and the bioinformatic cargo analysis. The reason was that, especially for bioinformatic cargo analysis, the amount of purified EVs needed was very high. These exceptions were specified in the text (Pag. 6 and pag. 15 line 401). Since cargo analysis didn’t result in enrichment of biomarkers relevant to our study, we chose to save 3D samples for the in vitro experiments. It is true that EVs from bioreactor are more homogeneous in size and pure, compared to 2D derived EVs. However both the two types of EVs were tested for cell-based functional assays and resulted in similar effects, meaning that also 2D-derived EVs can be used for small studies. On the contrary, bioreactor culture has to be preferred for industry manufacturing. Comment 5: Lines 113-116 Please specify range of pooled fractions. Response 5: Information has been added in the figure legend. Comment 6: Line 123 Purification runs were done in triplicate – include this statement in figure 2 also. Response 6: The statement has been added in the figure legend. Comment 7: Line 135 – swap upper case letters A and B to small letter case. Response 7: Change has been done, also for Fig. 4 legend. Comment 8: Lines 136-137 representative of three independent experiments – should be also stated in materials and methods section. Response 8: This information has been added also in Material and Methods section 4.5. Flow NanoAnalyzer analysis (Pag 17). Comment 9: Line 155 MVs – first time introduced abbreviation, include explanation for it. Response 9: Agree. Explanation of MV abbreviation has been added. Comment 10: Lines 158-159 – swap upper case letters A and B to small letter case. Response 10: In this case letters are matching in the figure and legend, thus we believe that it is not necessary to change them. Comment 11: Lines 177-184 – Specify here and in materials and methods section for western blot, which EVs sample is analyzed – from 2D or 3D culture. Response 11: Both samples have been analyzed, but there is an example of blot referring to 2D culture. This information has been added in the figure legend and Material and methods section 4.6. Western Blot pag. 17. Comment 12: Lines 202, 205, 206 etc. – uniform sample names throughout manuscript and in all figures/figure legends – i.e. SEC-sEVs or SEC-EVs. Response 12: Agree. We uniformed the nomenclature using SEC-sEVs and DG-UC-sEVs. When the term EV is intended as Extracellular Vesicles in general we used “EV”, but when we referred specifically to the samples characterized in the study we used “sEVs”. Comment 13: Line 208 – small proteins should be revised to either proteins or associated proteins, because larger proteins could also be contaminants, either by being soluble or associated as part of “soft” corona of EVs. Response 13: Agree. We revised the text. Comment 14: Line 242 – although authors stated it as limitations, they should explain why transcriptomic analysis was performed only with MSC-derived EVs purified from 2D CM. Response 14: As explained before, it was due to limited sample availability for these bioinformatic analysis. Comment 15: Line 282 – which MSC-EVs were used in these experiments? From 2D or 3D culture? Please specify here and in materials and methods section. Response 15: Both the samples were tested, this information has been specified in the text and in materials and methods section. Comment 16: Figures 15 and 17 – please change in figure DG-UC to DG-UC-EVs and SEC to SEC-EVs. Response 16: Figures have been modified. Comment 17: Figures 19 and 20 - DG-UC-EVs and SEC-EVs instead of DG-UC-sEVs and SEC-sEVs. Response 17: Figures have been modified. Comment 18: Supplementary Please revise “compared to the second sample” for clarification on what is second sample – “Table S1 reports the 31 proteins that are up-regulated in SEC- and DG-UC-EVs, respectively compared to the second sample”. Response 18: The statement has been revised. Comment 19: Discussion Lines 397-398 – include this reference too since it’s the first time anion exchange chromatography was used for purification of EVs [Kosanović M, Milutinović B, Goč S, Mitić N, Janković M. Ion-exchange chromatography purification of extracellular vesicles. Biotechniques. 2017 Aug 1;63(2):65-71. doi: 10.2144/000114575. PMID: 28803541.] Response 19: This new reference has been added. Comment 20: Authors could discuss about some differences in miRNAs between SEC-EVs and DG-UC-EVs, i.e. presence of other EVs in SEC-EVs sample which is more heterogeneous… Response 20: A statement expressing this concept has been added (line 413). Comment 21: Lines 409-410 – in results/materials and methods sections it is not specified which MSC-EVs were used, either from 2D or 3D culture, and figures results are only from one set of experiments 2D or 3D?). Response 21: This information has been added thoroughly in all the sections of the text. For the results section, figures have been presented as an example, the information about the sample tested has been specified in the figure legends. Comment 22: Lines 416-417 – “As a consequence of the higher purity of DG-UC-EVs”, is greater effect of DG-UC-EVs influenced by higher purity of EVs in sample in terms that there are less contaminants which may interfere or that greater heterogeneity of SEC-EVs, i.e. larger vesicles might also influence functional effects? – Very interesting topic that authors could also discuss in a sentence or two. Response 22: It is hard to make a conclusion on this. Both the two types of EVs exerted similar effects on cells, the only difference is that we needed more particles for SEC-EV samples to see a statistical significant difference compared to the control. Taking a dose of EV sample (calculated using the particle concentration) it could mean take more EVs in the case of purer samples (and less contaminants), or take a population of more heterogeneous EVs in the case of SEC-EVs which could have different and contrasting effects. Lines 428-432 explain this concept. Comment 23: Materials and methods Maybe in future experiments it would be more appropriate to centrifuge obtained CM at 300 x g before freezing in order to avoid possible intracellular vesicle contamination if some cells deteriorate during freeze-thaw cycle/cycles.
Line 441, 459 etc – uniform conditioned medium (CM) – when abbreviation is introduced first time, it should be used throughout the text Response 23: There was a mistake in the Material and Methods section, usually we collected CM and performed the pre-clearing centrifugation before freezing it at -80C. We corrected the material and methods paragraph 4.1, 4.2 and 4.3. Conditioned medium (CM) term has been uniformed in the text. |
Round 2
Reviewer 2 Report
Comments and Suggestions for Authors
Authors addressed all the points.
Accept the manuscript.
Author Response
Hello,
could you please let us know which of the 27 figures need improvement and which kind of improvement (resolution or other kind of editing)?
Thanks,
Marta Venturella